# Polydiacetylene-based ultrastrong bioorthogonal Raman probes for targeted live-cell Raman imaging

Sidan Tian[1], Haozheng Li[2], Zhong Li[3], Huajun Tang[2], Mingming Yin[1], Yage Chen[2], Shun Wang[4], Yuting Gao[1], Xiangliang Yang[1,5], Fanling Meng[1]*, Joseph W. Lauher[3]*, Ping Wang[1,2]* & Liang Luo [1,5]*

Live-cell Raman imaging based on bioorthogonal Raman probes with distinct signals in the cellular Raman-silent region (1800–2800 cm$^{-1}$) has attracted great interest in recent years. We report here a class of water-soluble and biocompatible polydiacetylenes with intrinsic ultrastrong alkyne Raman signals that locate in this region for organelle-targeting live-cell Raman imaging. Using a host-guest topochemical polymerization strategy, we have synthesized a water-soluble and functionalizable master polydiacetylene, namely poly(deca-4,6-diynedioic acid) (PDDA), which possesses significantly enhanced (up to ~10$^4$ fold) alkyne vibration compared to conventional alkyne Raman probes. In addition, PDDA can be used as a general platform for multi-functional ultrastrong Raman probes. We achieve high quality live-cell stimulated Raman scattering imaging on the basis of modified PDDA. The polydiacetylene-based Raman probes represent ultrastrong intrinsic Raman imaging agents in the Raman-silent region (without any Raman enhancer), and the flexible functionalization of this material holds great promise for its potential diverse applications.

[1] National Engineering Research Center for Nanomedicine, College of Life Science and Technology, Huazhong University of Science and Technology, Wuhan 430074, China. [2] Britton Chance Center for Biomedical Photonics, Wuhan National Laboratory for Optoelectronics, Huazhong University of Science and Technology, Wuhan, Hubei 430074, China. [3] Department of Chemistry, Stony Brook University, Stony Brook, NY 11790, USA. [4] MOE Key Laboratory of Fundamental Physical Quantities Measurement & Hubei Key Laboratory of Gravitation and Quantum Physics, PGMF and School of Physics, Huazhong University of Science and Technology, Wuhan 430074, P. R. China. [5] Hubei Key Laboratory of Bioinorganic Chemistry and Materia Medica, School of Chemistry and Chemical Engineering, Huazhong University of Science and Technology, Wuhan 430074, China. *email: fanlingmeng@hust.edu.cn; joseph.lauher@stonybrook.edu; p_wang@hust.edu.cn; liangluo@hust.edu.cn

Advances in optical microscopy in the past decades have played a key role in the development of modern biological sciences[1–4]. Raman scattering-based vibrational microscopy has been considered one of the most promising and powerful cell imaging tools in recent years[4–8], attributed to the many intriguing features it holds, including direct visualization of detailed molecular structure information, quantitative relationship between signal intensity and substance concentration, and narrow-band enabled multi-color imaging[9–13]. However, in comparison with other widely applied imaging techniques, especially fluorescence imaging, one of the major obstacles in practicing Raman microscopy in the biological sciences is the intrinsically poor cross-section of Raman scattering of either endogenous cellulous components or exogenous vibrational probes. Many techniques and strategies, such as surface-enhanced Raman scattering (SERS)[8,14–20], coherent anti-Stokes Raman scattering (CARS)[21–24], and stimulated Raman scattering (SRS)[9,25–29], have been developed to amplify the Raman signals of Raman probes or biomolecules. In particular, SRS imaging that can boost vibrational excitation represents a dramatic advance of detectability and imaging speed over conventional spontaneous Raman scattering. SRS also enjoys additional advantages, such as visualization of detailed distribution of biomolecules like lipids and proteins, pinhole-less three-dimensional optical sectioning, non-invasive observation and deep tissue penetration, and free from non-resonant background. However, to obtain effective SRS imaging in living cells requires coupling this optical imaging technique with intrinsic Raman-active molecular probes. Raman reporters with inherent strong Raman scattering intensity are still urgently needed to enable facile Raman imaging with high imaging quality.

Biological molecules in a cell, such as lipids, proteins, and nuclear acids are Raman active, exhibiting intense Raman signals in the regions of $400–1800 \, cm^{-1}$ and $2800–3100 \, cm^{-1}$. To efficiently eliminate interference of the endogenous cellular background, Bioorthogonal Raman probes with distinctive signals in the cell-silent vibrational region ($1800–2800 \, cm^{-1}$), including isotopes (C–D), azides, and triple bonds (C≡C and C≡N)-containing molecules, have been widely employed for labeled Raman imaging. Among the existing molecular Raman reporters, alkynes are the most promising candidates for live-cell imaging, because of their easy accessibility, minimal toxicity, and comparatively large Raman scattering cross-section. Since the initial demonstration of Raman bioimaging using 5-ethynyl-2′-deoxyuridine (EdU)[30], numerous alkyne-containing molecules have been exploited as bioorthogonal Raman probes for cell imaging by Raman microscopy[30–36]. However, the Raman signal intensity of C≡C bond stretching is still far from ideal and the detection sensitivity is very limited for the existing alkynes[27,34]. Polydiacetylenes, a class of conjugated polymers with an alternating ene-yne backbone structure, naturally emerge as promising Raman probe candidates, given their inherently ultrastrong Raman signals originating from the large π-conjugation and polarizability of the all-planar polymer backbones. However, most polydiacetylenes are insoluble in common solvents, since they are typically prepared via topological polymerization in solid state (Fig. 1a)[37–41]. In addition, to achieve successful polymerization, bulky side groups with strong inter-side chain interactions are often introduced[42]. The poor processibility and functionalizability of polydiacetylenes greatly limit their potential application in biological Raman imaging[43,44]. The host–guest supramolecular scaffolding strategy is a powerful tool for functional polydiacetylene synthesis[45]. With the help of the self-assembly of a host molecule, non-polymerizable diacetylene monomers can be properly aligned and successfully polymerized.

To address the above problems of polydiacetylenes, we have designed and synthesize a water-soluble and functionalizable polydiacetylene poly(deca-4,6-diynedioic acid) (PDDA) using the host–guest supramolecular scaffolding strategy (Fig. 1). Compare with conventional small molecule alkyne reporters, we observe an up to $\sim10^4$-fold enhancement of alkyne Raman signals of PDDA, making it an ultrastrong intrinsic alkyne Raman reporter. Figure 1b compiles the overlaid solution Raman spectra of PDDA and a variety of existing Raman probes. The concentration of C≡C bonds in the PDDA solution is several magnitudes lower than that of Raman-active bonds (C≡C, C≡N, and azide) in other probes, however, the C≡C bond peak intensity of PDDA is still considerably higher than that of any other one. More strikingly, PDDA is water soluble, and its propionic acid side chains allow facile and controllable side chain engineering of the polymer as a platform for a series of Raman probes (Fig. 1c). As a proof of concept, PDDA has been functionalized with subcellular organelle targeting groups, either small molecules or peptides (Fig. 1c). Based on these PDDA derivatives, we have achieved hyperspectral live-cell Raman imaging of subcellular organelles with high spatiotemporal resolution, using SRS at a low laser power and time constant.

## Results

**Synthesis of PDDA by host–guest topochemical polymerization.** We synthesized PDDA using the aforementioned host–guest topochemical polymerization strategy. The strong hydrogen bonding between carboxylic acids and Lewis bases provides an opportunity to create ordered crystals of the monomer deca-4,5-diynedioic acid (DDA), with a controlled molecular spacing. Our approach to the preparation of DDA co-crystals uses bis(pyridyl)oxalamide host **1**, which is composed of terminal Lewis basic pyridyl groups and a central oxalamide group that forms a self-complementary hydrogen bonding framework with the required spacing. The host–guest scaffold, as shown in Fig. 2a, can therefore align the DDA monomers as guests in the host framework with a spacing geometry that favors the topochemical polymerization.

The complete polymerization of DDA and generation of PDDA within the host–guest co-crystals were characterized by a single crystal X-ray diffraction experiment, as exhibited in Fig. 2b (side view) and Fig. 2c (top view). The crystal structure clearly demonstrates that the alkyne-containing polymer backbones are perfectly planar and parallel to the oxalamide hydrogen-bonding network, with a repeating distance of 4.82 Å, close to the ideal value of 4.9 Å for topochemical polymerization (Fig. 1a). The N–H distance of 1.76 Å and the N–H–O angle of 164° confirms that the host pyridines form the predicted hydrogen-bonds to the carboxylic acids. In addition, the high degree of polymerization and completely planar backbone conformation endow the polymer a metallic appearance and shiny gold color, as displayed in Fig. 2d.

Unlike most known polydiacetylenes that are insoluble in common solvents, the propionic acid substituents of the PDDA give the polymer high solubility in aqueous solutions. Taking advantage of this aqueous solubility, PDDA can be isolated from the host–guest co-crystals as a pure polymer by extensive acid-base rinsing. After being extracted from its crystalline lattice, the isolated polymer shows very good solubility in pure water and DMSO and interestingly, forms transparent aqueous solutions showing a remarkable chromism in the range of pH 6–8 (Fig. 2e and Supplementary Fig. 2a, b). At basic conditions (pH > 8), the PDDA water solution is yellow in color and has a maximum absorption of 460 nm. Density function theory (DFT) calculations suggests that the two adjacent substituents are perpendicular to each other in their ionized forms, therefore the conjugation of the backbone is minimized in a twisted conformation (Supplementary Fig. 2c). At lower pH values (pH < 6), the solution turns red

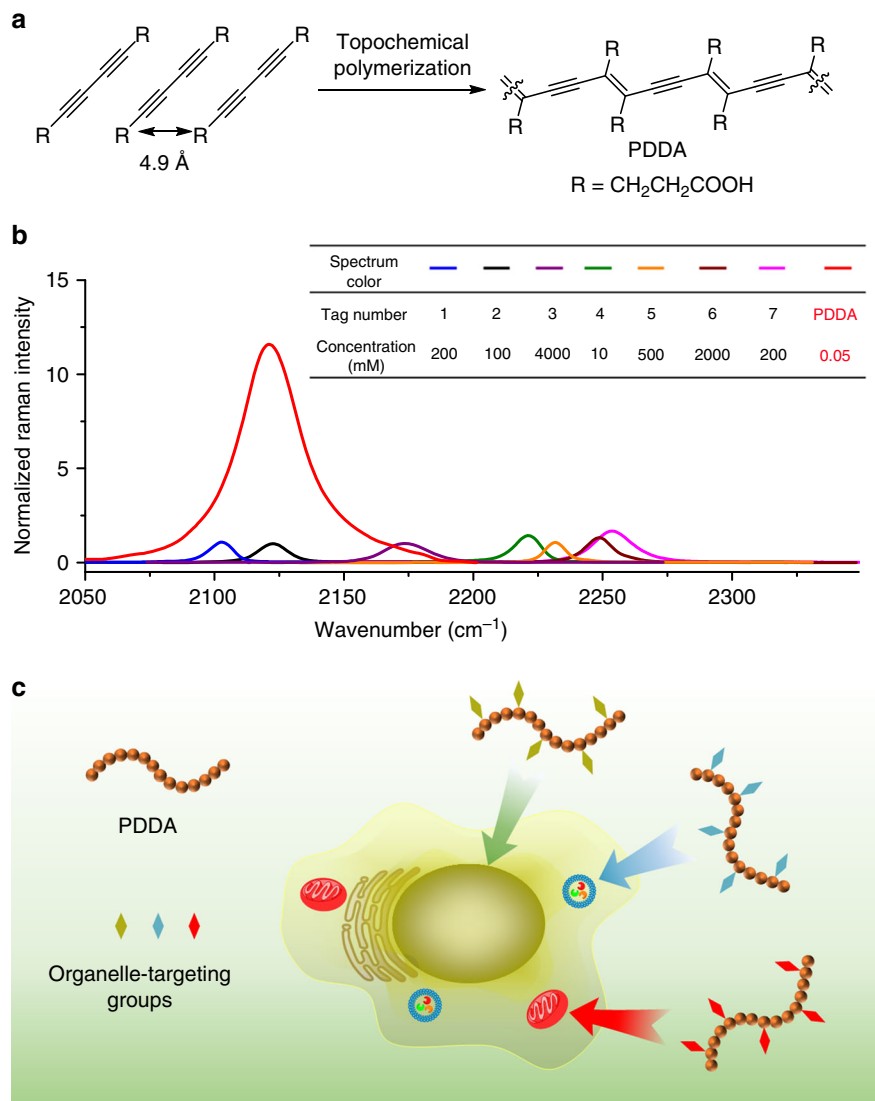

**Fig. 1 Polydiacetylene-based Raman probes for targeted live-cell Raman imaging. a** Topochemical polymerization for the preparation of PDDA. **b** Overlaid Raman spectra of individual DMSO solution of PDDA and a series of representative Raman probes, including 1. ethynylbenzene; 2. EdU; 3. diphenyl phosphorazidate; 4. diphenylbutadiyne; 5. benzonitrile; 6. 5-bromopentanenitrile; 7. deca-4,6-diynedioic acid. The probe structures are shown in Supplementary Fig. 1. The table inset lists the concentration of Raman-active bonds (C≡C, C≡N, or azide) in each solution. The normalization is based on the absolute Raman intensity of each spectrum. **c** Schematic illustration of side chain modification of PDDA for subcellular organelle targeting Raman imaging.

and the maximum absorption shifts to higher wavelengths. The red shift of the absorption corresponds to an increase of the backbone conjugation and planarity of PDDA, which may be induced by hydrogen bond-assisted inter-side chain assembly (Supplementary Fig. 2d).

**Comprehensive characterization of PDDA in solution.** The exceptional solubility of PDDA allows us to conduct a comprehensive exploration of the structure of PDDA and to assess the suitability of applying PDDA as a bioorthogonal Raman probe using solution-based methods, which has not yet been achieved by any other polydiacetylene. The Raman spectra of the polymer co-crystals and the polymer aqueous solution are similar, both displaying an intense peak corresponding to the C≡C bond stretches (2045 cm$^{-1}$ for co-crystals and 2120 cm$^{-1}$ for solution) in the cellular silent region (Fig. 3a). The C≡C bond Raman peak of the PDDA solution does not change upon irradiation under excitation with different wavelengths (785, 532, and 488 nm,

Supplementary Fig. 3). The solution-phase $^{1}$H NMR spectrum of the isolated polymer in D$_2$O shows only two broad proton peaks at 2.46 and 2.74 ppm, assigned to the $\alpha$ and $\beta$ methylene protons of the side chains of the polymer, Fig. 3b and Supplementary Fig. 4. The solution-phase $^{13}$C NMR spectrum of PDDA does not show the two acetylene peaks at 64.7 and 77.5 ppm found in the monomer spectrum. Instead, two new broad peaks at 99.9 and 129.8 ppm, corresponding to the $sp^2$-carbon and $sp$-carbon of PDDA, respectively, emerge from the polymer spectrum, unambiguously proving the successful topochemical synthesis and complete dissolution of the polymer. Gel permeation chromatography (GPC) with basic aqueous medium as the mobile phase (Supplementary Fig. 5) suggests that the prepared PDDA has a number-average molecular weight ($M_n$) of $2.4 \times 10^4$ g mol$^{-1}$. Its molecular weight distribution (PDI) of 1.45 is much narrower than other polydiacetylenes reported to date (Supplementary Table 1), suggesting the superior polymer quality achieved by the host–guest co-crystal strategy.

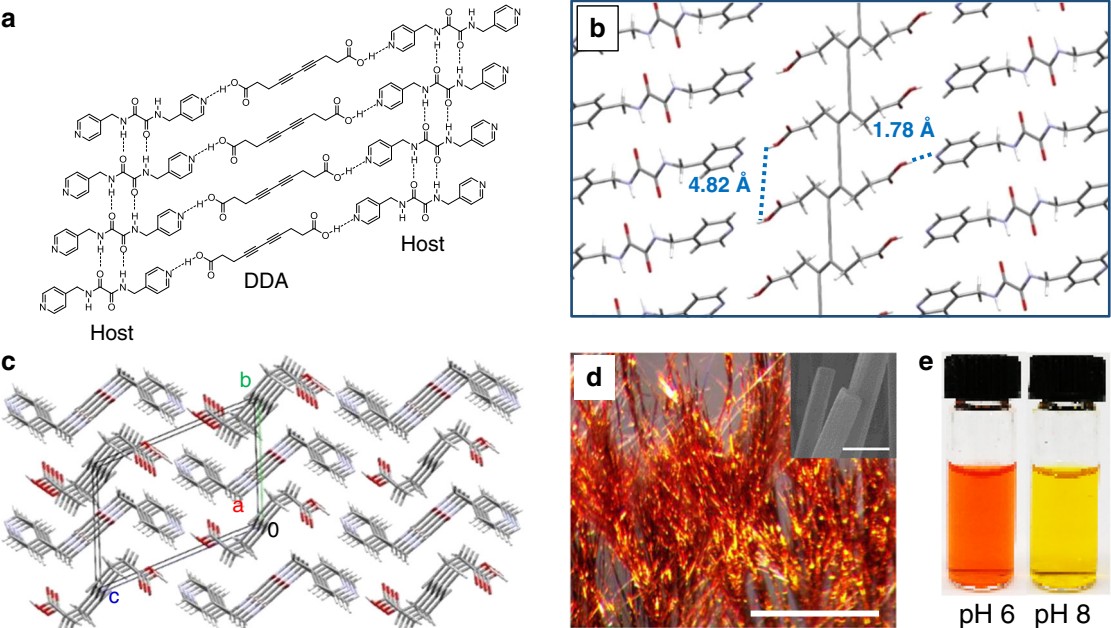

**Fig. 2 Topochemical polymerization within single crystals for the synthesis of PDDA. a** Ordered alignment of DDA monomers generated within a host–guest supramolecular framework with Host **1**. **b, c** Co-crystal structure of PDDA and host **1**, as determined by single crystal X-ray diffraction. **b** Side view; **c** Top view. The unit cell parameters (triclinic P-1) are a = 4.824(7) Å, b = 9.085(12) Å, c = 12.724 Å, α = 108.78(3)°, β = 96.71(3)°, γ = 96.19(3)°, V = 518.03 Å3, Z = 2. **d** Microscope image of PDDA·**1** co-crystals. Scale bar: 1 mm. Inset: Scanning electronic microscopy image of the corresponding co-crystals. Scale bar: 2 μm. **e** PDDA aqueous solution at different pH values.

**Ultrastrong Raman intensity of PDDA**. PDDA exhibits an exceptionally intense Raman peak at 2120 cm$^{-1}$ of PDDA in solution, resulting from the large polarizability of poly-diacetylenes[46,47]. In addition, the highly planar polymer backbone of PDDA further assists the stretching vibration of alkyne units in the backbone. To better calibrate the outstanding alkyne Raman intensity of PDDA, we normalized all the Raman spectra by C≡C bond concentration. The concentration of C≡C bonds in a PDDA solution was identical to the concentration of the repeating unit of PDDA. We used diphenylbutadiyne (DPY) as an internal standard to quantify the Raman intensity of PDDA, since the more commonly used EdU has a similar Raman peak at 2122 cm$^{-1}$ that overlaps with the alkyne peak of PDDA (~2120 cm$^{-1}$). DPY is a strong alkyne Raman tag, and its alkyne Raman peak (2220 cm$^{-1}$) can be clearly separated from that of PDDA. Figure 3c shows the Raman spectra of DMSO solutions of 50 mM DPY (100 mM C≡C bond) mixed with different con-centrations of PDDA (displayed as concentration of C≡C bonds). The Raman intensity of PDDA shows a perfect linear con-centration dependence (Fig. 3d), extending to when the C≡C bond concentration in PDDA solutions is as low as 1 μM, with a signal-to-noise ratio of over 10. Such a detection limit corre-sponds to a PDDA chain concentration of 8 nM, calculated based on the average polymerization degree of 120, which is far more sensitive than any other alkyne-containing molecule to date.

The Raman intensity of an alkyne-containing molecular Raman probe is typically evaluated by its relative Raman intensity versus EdU (RIE). We next calculated the RIE value of PDDA based on the intensities of the alkyne Raman peaks and the concentrations of the C≡C bonds of PDDA and DPY, using the below equation

$$\text{RIE}_{\text{PDDA}} = \text{RIE}_{\text{DPY}} \times \frac{I_{\text{PDDA}}/C_{\text{PDDA}}}{I_{\text{DPY}}/C_{\text{DPY}}} \qquad (1)$$

where $C$ is the alkyne concentration in each solution, $I$ is the alkyne Raman peak intensity of each solution. RIE$_{\text{DPY}}$ is the

relative alkyne Raman intensity of DPY versus EdU as literature reported[31]. We measured the RIE values of PDDA in DMSO solution at three different Raman excitation wavelengths (488, 532, and 785 nm), as shown in Supplementary Fig. 6a. The C≡C bond-normalized RIE value (RIE per C≡C bond, or RIE per repeating unit) were greater than 100 when measured at 785 nm Raman excitation. However, the C≡C bond-normalized RIE values exceeded 10$^4$ when we measured the Raman intensity with a Raman excitation wavelength of 488 nm or 532 nm. Since PDDA had a maximum absorption at 476 nm in DMSO (Supplementary Fig. 6b), the Raman excitation at 488 nm, a wavelength close to its absorption maximum, was able to further enhance the Raman intensity through pre-resonance Raman scattering[48,49], resulting in a tremendously high RIE value of 2.3 × 10$^4$ (C≡C bond-normalized).

We have also investigated how RIE value changes with the chromism of PDDA. RIE values of PDDA in a pH range from 5 to 9 were measured using a series of Raman excitation wavelengths (488, 532, and 785 nm). The results showed that the RIE values of PDDA exhibited a prominent increase with decreased pH values when excited by the 532 nm laser, consistent with the change of the absorption of PDDA at 532 nm (Supplementary Fig. 6c). The change of the RIE values of PDDA was much weaker when excited by the 488 nm or 785 nm laser, or by the 853 nm and 1040 nm Pump/Stokes laser beams for SRS imaging (Supplementary Fig. 7). As a comparison, the C≡C bond-normalized RIE value of the monomer DDA was measured to be ~0.85 regardless the Raman excitation wavelength (Supplementary Fig. 8), clearly evidencing the synergistic enhancement of the Raman intensity by both extension of conjugation length and pre-resonance Raman scattering.

Figure 4 summarizes the C≡C bond-normalized RIE values of PDDA (measured in DMSO by 488 nm excitation) and other typical alkyne-based Raman probes. As a comparison, the C≡C bond-normalized RIE values of another class of conjugated polymer poly(phenylene ethynylene) (PPE) are reported to be less

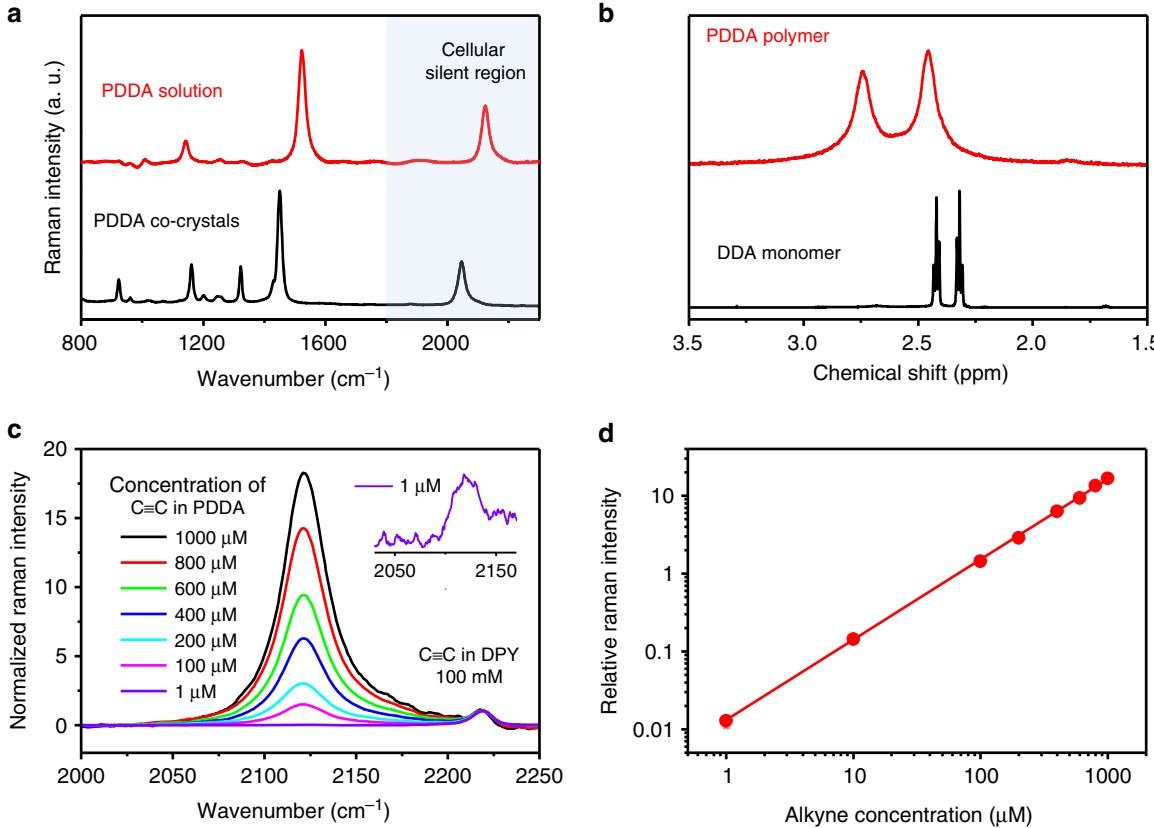

**Fig. 3 Characterization of PDDA polymer in solution. a** Raman spectra of PDDA aqueous solution and PDDA co-crystals. **b** Solution-phase $^1$H NMR spectra of PDDA polymer and DDA monomer (solvent: D$_2$O). **c** Raman spectra of mixed DMSO solutions of PDDA and DPY. The concentration of DPY in each solution is 50 mM (100 mM C≡C bond), and the calculated concentration of the C≡C bonds in PDDA is from 1 to 1000 μM. The Raman intensity of each spectrum is normalized to the DPY peak intensity. **d** Relative Raman intensity at 2120 cm$^{-1}$ of PDDA aqueous solutions plotted as a function of the concentration of C≡C bonds in PDDA. Solid line shows a linear fitting in the double logarithmic plot (R$^2$ = 1.000). Laser for Raman excitation $\lambda_{Ex}$ = 488 nm, 0.5 mW.

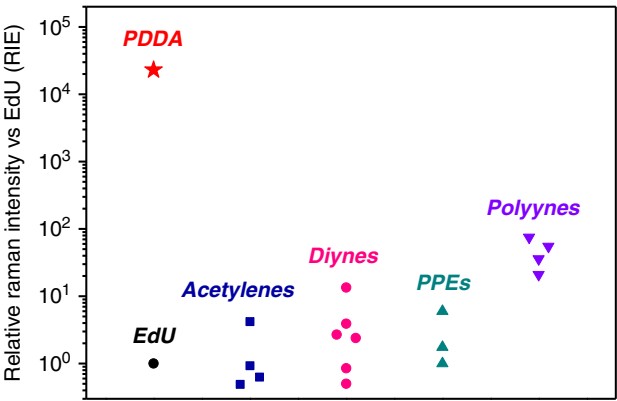

**Fig. 4 Plot of C≡C bond-normalized RIE values of various types of alkynes.** The detailed information of each data point, including RIE value, chemical structure, Raman shift, and resource, is available in Supplementary Table 2.

than 10$^{36}$. In addition, the Raman intensity of polyynes containing directly conjugated C≡C bonds also increases nonlinearly with the increase of the molecular length[28]. However, the synthesis of polyynes is highly difficult so that the length of polyynes is greatly limited[50]. The C≡C bond-normalized RIE value of PDDA is over 300 times higher than that of dodecahexayne, the longest polyyne used as a Raman probe,

demonstrating the enormous advantage of PDDA over other alkynes in terms of Raman intensity.

**PDDA-derivatives as super Raman probes**. The outstanding Raman sensitivity and the reactivity of propionic acid substituents of PDDA make it a potential precursor to a variety of bioorthogonal vibrational tags, through on-demand side chain modification using diverse biological targeting groups. We obtained three PDDA derivatives through modifications with different functional groups (Fig. 5), one with a tertiary amine (**P2**, for lysosome targeting) and the other two with targeting peptides (**P3** for mitochondria and **P4** for nucleus targeting), for subcellular organelle targeting in living cells. The tertiary amine was coupled directly to the carboxyl group on PDDA side chains following a typical condensation procedure, and we used Mal-PEG-NH$_2$ as a linker to conjugate PDDA with the two targeting peptides (Supplementary Fig. 9). FTIR spectra of the PDDA derivatives (Supplementary Fig. 10) displayed the newly formed amide bonds on the side chain of PDDA as well as PEG-related peaks. The significant red shifts of the absorption peaks of the PDDA derivatives (Supplementary Fig. 11) also suggested the formation of amide groups, which reduced electrostatic repulsion of the side chain and resulted in a more planar polymer backbone in solution through intramolecular hydrogen bonds. In addition, the Raman spectra of the PDDA derivatives (Supplementary Fig. 12) each still exhibited a strong C≡C bond peak at 2120 cm$^{-1}$,

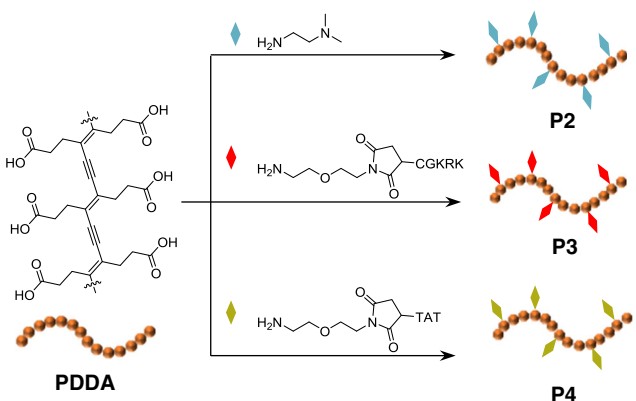

**Fig. 5 Side chain modification of PDDA with different functional groups to derivatives P2, P3, and P4 that can be targeted to specific subcellular organelles.** TAT: CRRRQRRKKR.

suggesting the retaining of the unique Raman features after side chain modification.

We next examined the targeting efficiency of the generated PDDA derivatives **P2**, **P3**, and **P4**. PDDA itself was not efficient in accumulating within live cells (Supplementary Fig. 13). However, the functionalized PDDA derivatives exhibited satisfactory performance in cell internalization and organelle localization. We used confocal laser scanning microscopy (CLSM) to image the intracellular distribution of the polymers within HeLa cells. After being co-incubated with standard fluorescence organelle tracking dyes, the fluorescence of the respective PDDA derivative overlapped precisely with that of the corresponding tracker (Supplementary Fig. 14), confirming the successful targeting of PDDA-derived tags to the specific sites. The cell uptake and targeted intracellular trafficking for PDDA-based macromolecular vibrational tags are therefore proved efficient and precise.

**Targeted Raman imaging in fixed and living cells.** To validate the superb Raman characteristics of PDDA-based vibrational tags, we performed hyperspectral SRS imaging of HeLa cells labeled with **P2**, **P3**, and **P4**. Figure 6a, b shows the schematic illustration of the hyperspectral SRS imaging system, which was implemented to perform molecular imaging in living cells. We used 853 nm pump laser beam and 1040 nm Stokes laser beam for the excitation of PDDA-based probes in the SRS imaging. After we specifically labeled lysosomes, mitochondria, and nuclei of fixed HeLa cells with these three different PDDA-based tags, hyperspectral SRS imaging was performed to illustrate the cellular distribution of the polymer tags in the cells using $2120\,cm^{-1}$ for the C≡C stretching. SRS imaging of HeLa cells treated with **P2** revealed accumulation of the tag in lysosomes of the cells (Fig. 6c, left). We obtained SRS spectra from the indicated region (red square) of the cell, and observed that the averaged SRS spectrum of **P2** in cells (circle dots in Fig. 6d) was almost identical to the spontaneous Raman spectrum of **P2** in solution (solid line in Fig. 6d). Moreover, SRS imaging of HeLa cells treated with **P3** displayed intense alkyne signals in the cytoplasm of cells which is consistent with the mitochondrial distribution (Fig. 6c, middle), while HeLa cells treated with **P4** exhibited dense alkyne signal accumulation in cell nuclei (Fig. 6c, right).

The extremely high Raman sensitivity of PDDA and its derivatives allows us to reduce the probe dose and apply ultralow laser power for SRS imaging, so as to minimize the interference and damage to the biological samples. As shown in

Supplementary Fig. 15, the cells stained with 50 μM of **P4** clearly showed a good SRS imaging at a very low Pump/Stokes laser power setting of 10/30 mW, i.e., 10 mW for the Pump laser and 30 mW for the Stokes laser, an integration time of 10 μs, and based on an average of 10 imaging frames. As a comparison, the cells stained with EdU remained undetectable at this condition, and the SRS imaging on these cells could only be obtained at a much higher power setting of laser beams (Pump/Stokes 50/100 mW), a longer integration time of 40 μs, and based on a larger number of 50 imaging frames for averaging. This parallel SRS imaging study on cells stained with EdU and **P4** unambiguously demonstrated how PDDA-based probes outperformed typical existing probes for SRS.

Figure 7 illustrated hyperspectral SRS imaging of living HeLa cells stained with the above PDDA-derived tags (**P2**, **P3**, and **P4**) at Pump/Stokes 10/30 mW, integration time of 40 μs, and with an average of 10 frames. The cellular distribution of the polymer tags and inherent lipids and proteins of cells were collected using $2120\,cm^{-1}$ for the C≡C stretching and $2850\,cm^{-1}$ for the C–H vibrations. The $2120\,cm^{-1}$ channel of the SRS images of living HeLa cells treated with different polymer tags clearly visualized the accumulation of the corresponding tag in lysosomes (**P2**, Fig. 7, first row), mitochondria (**P3**, Fig. 7, second row), and nuclei (**P4**, Fig. 7, third row) of the cells, respectively. In addition, we used commercial fluorescence organelle trackers to co-incubate PDDA-based probes in corresponding cells, and conducted SRS imaging of PDDA-based probes and two-photon fluorescence imaging (TPFI) of commercial organelle trackers on the same instrument. The parallel comparison between the TPFI images of the commercial trackers and the SRS images of the PDDA-based probes (Supplementary Fig. 16) clearly evidences the specificity of these conjugated probes to the desired targets.

On the other hand, the $2850\,cm^{-1}$ channel for the C–H vibrations clearly illustrated the healthy cell morphology and nuclear shape. The localization information of targeted organelles, together with the distribution of endogenic biomolecules such as lipids and proteins, provided a comprehensive cellular information by SRS imaging. To further confirm the biocompatibility of these probes, we studied the cytotoxicity of all PDDA-based probes using MTT assays (Supplementary Fig. 17). When the HeLa cells were treated with 50 μM of **P2**, **P3**, or **P4**, the cell viabilities remained above 90% for all three probes after 48 h of treatment. It should be noticed that when stained with **P2** or **P3**, the cells just needed to be incubated with the probe (50 μM) for less than 6 h, and staining cells with **P4** took a relatively longer treatment time of 48 h. In a more stressful condition, we treated the cells with 100 μM of PDDA-based probes, the cell viabilities after 48 h of incubation were all above 80%. The MTT experiment results clearly evidenced the biocompatibility of the PDDA-based Raman probes, and they are safe and applicable for Raman imaging of live cells.

In summary, we demonstrate that polydiacetylene derivatives with ultrastrong alkyne Raman signals can be used as intrinsic Raman reporters for live-cell Raman imaging. The C≡C bond-normalized RIE value of PDDA can reach $2.3 \times 10^4$, which is up to $10^4$ fold higher than other existing alkynes. Functionalizing side chains of PDDA endows us a wide variety of polydiacetylene-based super vibrational tags that are advantageous as Raman-active nanomaterials for distinct imaging application. In addition, PDDA can be used in conjunction with other Raman enhancing techniques, such as SERS, to achieve even stronger Raman-active nanomaterials that are expected to break the existing sensitivity ceiling of Raman imaging. Developing PDDA-based multiplex Raman probes, as well as exploiting more applications of Raman imaging on the basis of PDDA derivatives is currently on the way.

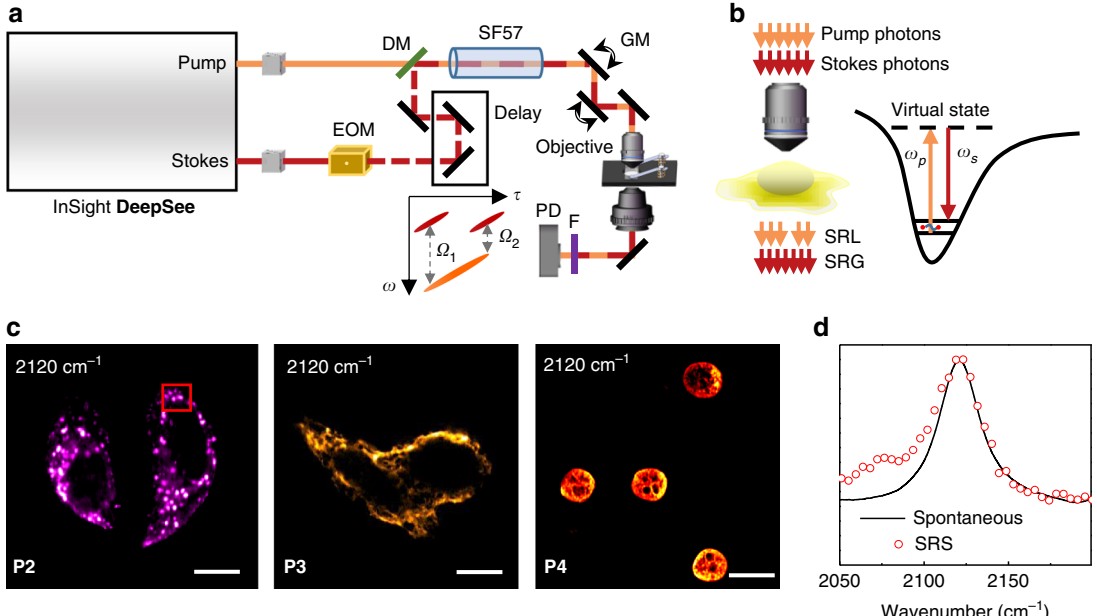

**Fig. 6 Illustration of the SRS system and SRS images of fixed HeLa cells. a** Schematic illustration of the instrumental setup of the SRS imaging system. EOM: electro-optic modulator; DM: dichroic mirror; GM: galvanometer; F: optical filters; PD: photodiode. **b** Stimulated Raman transition process for vibrational imaging of HeLa cells stained by PDDA-based tags. **c** SRS images (2120 cm$^{-1}$) of fixed HeLa cells treated with 50 μM of **P2**, **P3**, and **P4**, respectively. Scale bar: 10 μm. **d** SRS spectrum of the HeLa cell sample stained with **P2** in the red box region in (**c**), the spontaneous Raman spectrum of **P2** in aqueous solution shown as a reference.

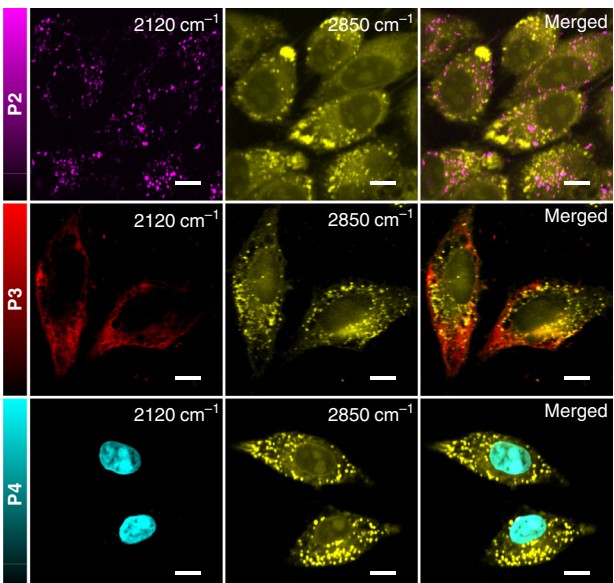

**Fig. 7 SRS images of HeLa cells treated with 50 μM of P2, P3, and P4, respectively.** Images shown from left to right are the alkyne (2120 cm$^{-1}$), lipids (2850 cm$^{-1}$), and merged images. Scale bar: 10 μm.

## Methods

**Preparation of PDDA**. In a typical run, 27.0 mg of host **1** was dissolved in 30 mL methanol with sonication to give a colorless solution. 19.4 mg of DDA was then added to the above solution and the mixture was centrifuged at 2400 × *g* for 10 min to remove possible dust impurities. The resulting solution was transferred to a crystallization dish and kept in 4 °C. The slow evaporation of the solvent generated white needle-like crystals in the crystallization dish. Heating the crystals in an oven set at 120 °C for 12 h turned the color of the crystals from pink to dark red then metallic gold. The final product was characterized with X-ray single crystal diffraction and Raman spectroscopy, which confirmed that the gold-colored crystals were PDDA-host **1** co-crystals. The crystallographic data have been deposited in the Cambridge Crystallographic Data Center (CCDC) with an accession number of CCDC: 1900689.

**X-ray single crystal diffraction**. Crystals were selected and mounted on glass fibers using epoxy glue. The crystals were optically centered on a Bruker AXS SMART CCD diffractometer and diffraction data were collected using a Siemens graphite-monochromated Mo radiation tube. The unit cells were determined by a least-squares analysis using the SMART software package. The structures were solved and refined with standard SHELX procedures.

**PDDA isolation**. Dispersing the polymerized co-crystals in 0.1 M aqueous NaOH and extensive rinsing for 1 h yielded a yellow suspension, followed by removing undissolved host **1** by filtration. The basic PDDA solution was then acidified with hydrochloric acid to pH 1. The solution turned cloudy and was kept at 4 °C overnight. Red solid precipitated from the solution, which was collected and washed with dilute hydrochloric acid and methanol. Drying the solid in vacuum at room temperature gave dark red PDDA products. $^1$H NMR (400 MHz, D$_2$O) Chemical shift (ppm) 2.74 (br, CH$_2$), 2.46 (br, CH$_2$). $^{13}$C NMR (100 MHz, D$_2$O) Chemical shift (ppm) 182.5, 129.8, 99.9, 36.8, 32.3.

**Nuclear magnetic resonance (NMR)**. All NMR spectra were recorded on an Agilent 400-MR 400 MHz spectrometer operated in the Fourier transform mode. Methanol-d$_4$ and D$_2$O were used as solvents. In case the PDDA polymer has a low solubility in D$_2$O at neutral or acidic pH, the sample in D$_2$O is basified with sodium carbonate to achieve a satisfactory polymer concentration for NMR measurements.

**PDDA side chain modification**. P2: 1 mg of PDDA was dissolved in 1 mL PBS (1× pH 7.4). NHSS (2.17 mg) and EDCI (3.84 mg) were dissolved in 0.1 mL PBS (1× pH 7.4), and added to the PDDA solution. After the mixture had been gently shaken at room temperature for 2 h, the N,N-diethylethylenediamine (12 mg) in 0.1 ml PBS (1× pH 7.4) was added to the solution for conjugation. After being stirred at room temperature for 12 h, the reaction solution was placed in a dialysis bag (intercept molecular weight 3500 Da) against basic water (pH 10) for 8 h and then pure water for additional 48 h. Lyophilization of the resulting polymer solution generated **P2**.

P3 and P4: In a typical run, PDDA reacted firstly with freshly prepared Mal-PEG$_{2000}$-NH$_2$, following a procedure identical to the N,N-diethylethylenediamine reaction. 10 mg Mal-PEG$_{2000}$ modified PDDA was dissolved in 1 mL PBS (1× pH 7.4) and 10 μmol thiol group containing peptide CGKRK or TAT is added to the solution. After reacting at room temperature for 12 h, the reaction mixture was loaded into a dialysis bag (intercept molecular weight 7000 Da) and dialyzed against pure water for 48 h. The resulting polymer solution was diluted to a fixed alkyne concentration and iso-osmotically adjusted with 20 × PBS before being used in living cell Raman tests.

**Spontaneous Raman spectroscopy**. Raman detection unit composed of a spectrometer (Princeton Instruments Acton series SP-2500) and a liquid nitrogen cooled CCD camera (Princeton Instruments PyLoN-100BR-eXcelon) is used for Raman spectra measurements. A Sapphire SF 488 nm laser is applied to the sample with 2 mW laser power.

**Relative Raman intensity measurements**. 10 μL of the mixed solution of PDDA and DPY was sealed on a glass slide and then measured with a confocal Raman spectrometer. Optimized signal acquisition conditions for the DPY standard were applied. The peaks intensities of alkyne in PDDA and DPY were used to evaluate the relative Raman intensity. $RIE_{PDDA}$ could be calculated with Eq. (1). As we observed a linear relationship between relative $I_{PDDA}$ with $C_{PDDA}$ using DPY with fixed concentration as a standard, the above Eq. (1) could be simplified as

$$RIE_{PDDA} = RIE_{DPY} \times slope \qquad (2)$$

All concentrations were quantified by the alkynes. The RIE values of the molecules containing multiple C≡C bonds were divided by the number of C≡C bonds prior to the comparison.

**Cell culture for imaging**. HeLa cancer cell lines were purchased from the American Type Culture Collection (ATCC) and cultured in Dulbecco's modified Eagle's medium (DMEM) with 10% (v/v) fetal bovine serum (FBS) and antibiotics (penicillin/streptomycin). For SRS imaging experiments, PDDA-based Raman probes were incubated with cells in a glass-bottom petri dish (diameter 3.5 cm). Freshly prepared **P2**, **P3**, and **P4** were added to 1.5 mL DMEM culture medium containing 10% FBS and antibiotics (penicillin/streptomycin) to reach a final concentration of 50 μM (concentration based on C≡C in PDDA) and the used as a working solution. For lysosome-staining, the cells were incubated with the **P2**-containing culture medium for 2 h and then washed with 1× PBS for 3 times before fresh culture medium was added. For confocal fluorescence imaging or two photon fluorescence imaging, Lyso-tracker Red was then added to stain the cell following a standard protocol. The cells were then washed with 1× PBS for 3 times and 1 mL 1× PBS was added to immersed the cells for imaging. Similar procedures were conducted for **P3** to stain mitochondrion and **P4** to stain nucleus. The incubation time was 6 h for **P3** and 48 h for **P4** to ensure enough accumulation in the corresponding organelles. For fixed cell imaging, 4% Paraformaldehyde in 1× PBS was added to the cell samples after the staining procedure. After 10 min of incubation at 37 °C, the sample was washed by 1× PBS for 3 times and 1 mL 1× PBS was added to immerse the cells for imaging.

**Stimulated Raman scattering imaging**. The dual-output femtosecond laser (InSight DeepSee, Spectra-Physics) provided both pump (680–1300 nm, ~120 fs) and Stokes (1040 nm, ~220 fs) laser beams with a repetition rate of 80 MHz. The Stokes beam was modulated by a resonant electro-optical modulator (EO-AM-R-C2, Thorlabs) at 10.55 MHz with modulation depth about 95%. The temporal overlap between pump and Stokes beams was ensured with a time delay line. The pump beam was spatially overlapped with Stokes beam by a dichroic mirror (DMSP1000L, Thorlabs). Pump and Stokes pulses were linearly chirped to ~3 ps by 64 cm long SF57 glass rod. The hyperspectral image stack was obtained by scanning the relative time delay between pump and Stokes pulses. The two laser beams were guided into a laser scanning microscope equipped with a two-axis galvanometer (GVS002, Thorlabs). A ×60 water immersion objective (N.A. 1.1, LUMFLN 60XW, Olympus) was used for all cell imaging, and the transmitted pump beam was collected with a high N.A. oil condenser and detected by a large area Si photodiode (S3994-01, Hamamatsu). Two shortpass filters (ET980SP, Chroma) were installed in front of the photodiode to completely block the Stokes beam. The SRS signal detected by PD was amplified by an in-house built 10.5 MHz resonant amplifier and demodulated by a digital lock-in amplifier (LIA, HF2LI, Zurich Instrument). The signal integration time for a pixel of 250 nm × 250 nm was 10 μs, the typical acquisition time for each frame with an area of 100 μm × 100 μm was around 1.6 s. To further increase the signal-to-noise ratio of the image, we averaged the results of 10 frames for the final image and the total time for each image in the manuscript was less than 30 s.

## Data availability

The X-ray crystallographic coordinates for structures reported in this study have been deposited at the Cambridge Crystallographic Data Centre (CCDC), under a deposition number 1900689. These data can be obtained free of charge from The Cambridge Crystallographic Data Centre via www.ccdc.cam.ac.uk/data_request/cif. All relevant data are available within the Article, Supplementary Information, Source Data file or available from the authors upon reasonable request.

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

## Acknowledgements

This work is supported by the National Basic Research Plan of China (2018YFA0208903), the National Natural Science Foundation of China (21877042 and 21702070), Postdoctoral Research Foundation of China (2017M612461 and 2017M622454), the National Science Foundation (Grant CHE-0453334), and Huazhong University Startup Fund. P.W. acknowledges the supports from the National Natural Science Foundation of China (61675075), the National Key Research and Development Program of China (2016YFA0201403), Science Fund for Creative Research Group of China (61421064). We also thank the Analytical and Testing Center of Huazhong University of Science and Technology for related analysis.

## Author contributions

S.T., H.L., and Z.L. contributed equally. S.T., H.L., and Z.L. designed and performed the experiments. H.T and Y.C. participated some of the SRS work. M.Y. and Y.G. assisted with some of the synthesis and crystallization work. X.Y., S.W., and F.M designed the experiments. J.W.L., P.W., and L.L. conceived and obtained funding for the project, oversaw the research and wrote the paper. All authors discussed the results and commented on the paper.

## Competing interests

The authors declare no competing interests.
