## [Peer Review File · Nature Communications]

Reviewers' Comments:

Reviewer #1:

Remarks to the Author:

Since alkyne-tag Raman imaging was developed in 2011, the application range of Raman imaging has been expanding in recent years. However, Raman imaging is currently far inferior in sensitivity to fluorescence imaging. Therefore, developing a probe that exhibits strong Raman scattering is a very important issue. The authors have succeeded in developing an ultra-strong Raman probe PDDA with a RIE value of 23,000 per alkyne unit. In addition, the authors have modified PDDA to develop Raman-active nanomaterials that selectively accumulate intracellular organelles. These results are significant as basic research, but in terms of application, it has not become possible to do what fluorescence imaging or Raman imaging could ever do. Therefore, the impact of the content is not sufficient for the articles published in Nature Communications. In addition, discussions on the following points are insufficient.

1. Since PDDA has an absorption maximum close to the excitation wavelength, it is considered that the contribution of the pre-resonance Raman scattering is large. It is necessary to measure and compare the RIE values at several wavelengths to indicate the magnitude of the contribution.
2. It should be discussed what structural change the chromism shown in Fig S2 is derived from. Since the carboxylic acid and poly-yne are not conjugated, is there any structural change occurring with deprotonation?
3. In cells, pH varies depending on organelles. In order to perform cell imaging, it is necessary to show how RIE value changes due to chromism.
4. In the imaging of Fig. 7, all of the lipid, mitochondria, and nucleus are in a damaged shape. In addition, the treatment time is not described in the MTT test. Current data is insufficient to show that it is applicable to live cell imaging.
5. Since the probe shows strong Raman scattering, it is more meaningful to show the result of spontaneous Raman imaging instead of nonlinear Raman imaging.
6. Strategies for polymerization are not uncommon. Appropriate articles should be cited.
7. The name of EdU described in line 59 is incorrect. Correctly, 5-Ethynyl-2'-deoxyuridine. Also, the structure of EdU described in Fig S1 lacks double bonds.

Reviewer #2:

Remarks to the Author:

Overall the paper is well written and relatively easy to understand. The strongest part of the paper is clearly the thorough characterization of the chemistry of the PDDA Raman probes. The authors did a lot of work characterizing the probes and used several techniques to validate their findings including, NMR, x-ray crystal diffraction, Raman spectroscopy, Gel Permeation Chromatography, and UV/Vis Spectroscopy.

It would be valuable to the reader to understand what the overall applications for using SRS imaging and this new probe could be and why it could be important to scientists, especially for this journal that can have a broader audience.

The authors mention in their first sentence that "Bioorthogonal Raman imaging agents with enhanced Raman signals in the cellular Raman-silent region (1800–2800 cm⁻¹) are highly desired for effective Raman imaging of living cells." But then leave the reader with no explanation as to

why.

Also, how is this technique better than what is already conventionally used for instance fluorescence imaging of live cells? A little more on the applications of this new Raman probe would be helpful.

The biggest weakness of this paper is the cell studies. The Raman images generated aren't completely convincing with regards to the specificity of these conjugated probes to the desired targets. The fluorescence images in S9 look a lot more convincing than the Raman images in the end.

There also needs to be more details on how long the SRS imaging took in terms of acquisition time. It would also be nice to see a control image of what an untargeted PDDA looks like after applied to the cells, to look at non-specific accumulation of the PDDA, you can't assume that the small molecule probe won't internalize passively: "PDDA itself would not be expected to accumulate within cells, given its polyanionic nature in physiological environment".

In Figure 7, it is unclear what the lipid channel is highlighting, just the outline of the cell? The P2 images don't look terribly impressive, why are there artifacts outside the cell region? There are also what appears to be lysosomes in the nuclear regions? And the nuclear targeting as stated in your paper is also "off" due to potential cytoplasm-nucleus translocation...but that's not good for live cell imaging applications if you can't get the real image of the nuclear component. Is there a way to validate the SRS images showing the presumed targeting of the PDDA to your targets with fluorescence images at the same time? To overlay their distribution within the cell? Again, it would be nice to know how long it took to generate these SRS images.

Clearly the PDDA has a nice peak in the "silent region", and it seems to outperform many of the pre-existing probes, but using it in this live cell imaging application doesn't seem all that convincing as of yet. I believe more details on how it outperforms conventional live cell imaging (ie. fluorescence) and more cellular imaging experiments are necessary before publication under the current title "Polydiacetylene-Based Ultrastrong Bioorthogonal Raman Probes for Targeted Live-Cell Raman Imaging".

Minor: The last sentence of the conclusion also needs to be restructured grammatically.

Again the chemistry and characterization are well demonstrated for the new Raman probe but its application for live cell imaging needs to be further investigated.

Reviewer #1:

We really appreciate that the reviewer considered our results significant as basic research, and we also thank the reviewer for raising substantial concerns about the application of these materials. We have conducted a series of additional experiments to demonstrate the potential of polydiacetylene-based Raman probes in the application of live-cell Raman imaging. Below please find our point-by-point response to the critiques by Reviewer 1.

- 1. Since PDDA has an absorption maximum close to the excitation wavelength, it is considered that the contribution of the pre-resonance Raman scattering is large. It is necessary to measure and compare the RIE values at several wavelengths to indicate the magnitude of the contribution.**

Response: We thank the reviewer for this very valuable advice. We totally agree that the contribution of the pre-resonance Raman scattering is large in boosting the Raman signals of the PDDA-based Raman probes. In fact, the ultrastrong Raman intensity of PDDA is a result of synergistic enhancement of both topochemical polymerization and pre-resonance Raman scattering. As the reviewer suggested, we measured the RIE values of PDDA at three different Raman excitation wavelengths (488 nm, 532 nm, and 785 nm), as shown in Figure R1A and Supplementary Fig. 6A. The C≡C bond-normalized RIE value were greater than 100 when measured at 785 nm. However, the C≡C bond-normalized RIE values exceeded 10^4 when we measured the Raman intensity with a Raman excitation wavelength of 488 nm or 532 nm. Since PDDA had a maximum absorption at 476 nm (Figure R1B and Supplementary Fig. 6B), the Raman excitation at 488 nm, a wavelength close to its absorption maximum, was able to further enhance the Raman intensity through pre-resonance Raman scattering,^{1,2} resulting in a tremendously high RIE value of 2.3×10^4 (C≡C bond-normalized). As a comparison, the value of the monomer DDA was measured to be ~ 0.85 regardless Raman excitation wavelength (Figure R1C and Supplementary Fig. 8), clearly evidencing the synergistic enhancement of the Raman intensity by both extension of conjugation length and pre-resonance Raman

scattering. We have revised the manuscript and added related discussion in Page 13 (highlighted in red). We have also updated Supplementary Fig. 6, Fig.8 with added RIE values at different excitation wavelengths.

Figure R1. A) RIE values of PDDA in DMSO measured with different Raman excitation wavelengths (488 nm, 532 nm, and 785 nm). B) UV/Vis absorption spectrum of a DMSO solution of PDDA. C) RIE values of DDA in DMSO measured with different Raman excitation wavelengths (488 nm, 532 nm, and 785 nm).

2. It should be discussed what structural change the chromism shown in Fig S2 is derived from. Since the carboxylic acid and poly-yne are not conjugated, is there any structural change occurring with deprotonation?

Response: Thank the reviewer for the question. Chromism of conjugated polymers, including polydiacetylenes, is mainly derived from the change of conjugation length and planarity of the conjugated backbone.³⁻⁵ In PDDA, it is true that the carboxylic acid side group and the polymer backbone are not directly conjugated, instead they are linked by two methylene groups. However, when the carboxylic acid side groups are protonated, they can form hydrogen bonds between the adjacent carboxylic acid groups, which helps to maintain the planarity of the conjugated polymer backbone. Therefore, the maximum absorption of the polymer in acid condition is relatively larger. On the contrary, when the carboxylic acid side groups are deprotonated, the negative charges on the terminal side chain groups exhibit strong electrostatic repulsions and the planarity of the backbone is disrupted accordingly. As a result, the conjugation of the backbone is minimized and the maximum absorption shifts to lower wavelengths. In addition, we calculated the conformational change of a PDDA fragment with 6 repeating units in the protonated and deprotonated forms using density functional theory (DFT). As shown in Figure R2, the calculation results suggested that the backbone of PDDA was planar in the protonated form. While in the ionized form, the two adjacent substituents were perpendicular to each other, leading to a twisted conformation of the backbone. The conformational change of PDDA backbone induced by the protonation/deprotonation process was in coincidence with the pH-dependent chromism phenomenon shown in Supplementary Fig. 2. We have revised the manuscript (highlighted in red in Page 9) and updated Supplementary Fig. 2 accordingly.

Protonated

Deprotonated

Figure R2. Density functional theory (DFT) calculation with b3lyp/6-31g(d) method of the conformations of a PDDA fragment (6 repeating units) with its carboxyl groups in protonated and deprotonated forms.

3. In cells, pH varies depending on organelles. In order to perform cell imaging, it is necessary to show how RIE value changes due to chromism.

Response: Thank the reviewer for this very important suggestion. We have measured different RIE values of PDDA from pH 5 to pH 9, using a series of Raman excitation wavelengths (488 nm, 532 nm, and 785 nm), to investigate how RIE value changes due to chromism. The results showed that the RIE values of PDDA exhibited a prominent increase with decreased pH values when excited by the 532 nm laser, consistent with the change of the absorption of PDDA at 532 nm (Figure R3B, Supplementary Fig. 6D in the Supplementary Information). The change of the RIE values of PDDA was much weaker when excited by the 488 nm or 785 nm laser, or by the 853 nm and 1040 nm Pump/Stokes laser beams for SRS imaging (Figure R3C and Supplementary Fig. 7 in the Supplementary Information). We have added this discussion in the manuscript on Page 13, and updated Supplementary

Fig. 6, Fig. 7 accordingly.

Figure R3. A) pH-dependent RIE values of PDDA measured at different Raman excitation wavelengths (488 nm, 532 nm, and 785 nm). B) pH dependent absorption spectra of PDDA. C) RIE values of PDDA as a function of pH values measured by SRS (excitation: 853 nm; pump: 1040 nm).

4. In the imaging of Fig. 7, all of the lipid, mitochondria, and nucleus are in a damaged shape. In addition, the treatment time is not described in the MTT test. Current data is insufficient to show that it is applicable to live cell imaging.

Response: We really appreciate this critique. We agree that the quality of the SRS

imaging of cells labeled by PDDA-based Raman probes was not satisfactory and the organelles were not in good shapes. In our previous imaging experiments, to avoid possible interference of serum, we used fetal bovine serum (FBS)-free culture medium for the live cell staining. However, this resulted in unhealthy morphology of the cells, especially for the nucleus-targeting probe **P4** which needed a long incubation time up to 48 h. We then tried to conduct the staining procedure in DMEM medium containing 10% FBS. As shown in Figure R4 (updated Figure 7 in the manuscript), the cell morphology was significantly improved, and the lipid droplets, mitochondria, and the cell shape were all in healthy shapes. Even for the nucleus-targeting probe that had been incubated for 48 h, the shapes of the nuclei as well as the cells were maintained well. The successful SRS imaging results indicate that the PDDA-based Raman probes are resistant to the interference of serum and have good potential to be applied in complicated physiological environments. We have replaced the SRS images in Figure 7 with newly obtained results and changed the description in the Supplementary Information accordingly.

Figure R4. SRS images of HeLa cells treated with 50 μM of **P2**, **P3**, and **P4**, respectively. Images shown from left to right are the alkyne (2120 cm^{-1}), lipids (2850 cm^{-1}), and merged images. Scale bar: 10 μm .

In addition, we apologize that we did not mention the treatment time of the Raman probes in the MTT test. In our initial manuscript, the MTT experiments were carried out after the cells were incubated with PDDA derivatives for 48 h. To systematically evaluate the cytotoxicity of PDDA-based Raman probes, we performed the MTT tests on cells with different treatment time. As shown in updated Figure R5 (Supplementary Fig. 16), When the HeLa cells were treated with 50 μM of **P2**, **P3**, or **P4**, the cell viabilities remained above 90% for all three probes after 48 h of treatment. It should be noticed that 50 μM was the highest concentration we used for cell imaging. In addition, only **P4** needed a treatment time for 48 h. When stained with **P2** or **P3**, the cells just needed to be incubated together with the probe for less than 6 h. In a more stressful condition, we treated the cells with 100 μM of PDDA-based probes, the cell viabilities after 48 h incubation were all above 80%. The MTT experiment results clearly evidenced the

biocompatibility of the PDDA-based Raman probes, and they are safe and applicable for Raman imaging of live cells. We have updated Supplementary Fig. 16 with the new data.

Figure R5. MTT assays of PDDA derivatives **P2**, **P3**, and **P4** with different concentrations and incubation time.

5. **Since the probe shows strong Raman scattering, it is more meaningful to show the result of spontaneous Raman imaging instead of nonlinear Raman imaging.**

Response: Thank the reviewer for the kind advice. We understand that it should be more reasonable to conduct spontaneous Raman imaging since PDDA-based probes exhibit strong Raman scattering. In this manuscript, we demonstrated the SRS imaging of PDDA-based probes mainly because of a few advantages of SRS over spontaneous Raman imaging. First, SRS imaging retains most unique characteristics Raman imaging holds in general, such as direct visualization of detailed molecular structure information, quantitative relationship between signal intensity and substance concentration, and narrow-band of the signals. In addition, compared with spontaneous Raman imaging in which the sample is illuminated with one optical beam, SRS utilizes two beams at the pump frequency (shorter wavelength beam) and Stokes frequency (longer wavelength beam) to coincide with

the sample, which brings in a number of additional advantages, including pinhole-less three-dimensional optical sectioning, non-invasive observation, and free from non-resonant background.

More importantly, we choose SRS imaging also because this technique is much faster and convenient than spontaneous Raman imaging. Most available spontaneous Raman confocal microscopes use motor-driven sample stages to achieve Raman mapping, and the average time to capture a single image is several hours, which is too long for live cell imaging. In addition, real-time focal plane adjustment is almost impossible for spontaneous Raman scattering. As for the SRS technique, scanning galvanometer-controlled laser enables a fast imaging of live cells within a few seconds. In addition, the excitation wavelength of SRS imaging is usually in the near infrared region, which allows less photo damage to live cells and deeper tissue penetration, which will be ideal for future in vivo applications. If we demonstrate successful SRS imaging of live cells by PDDA-based probes, we will certainly move forward to achieve targeted in vivo SRS imaging using these probes, which seems unlikely to be benefited from spontaneous Raman imaging.

Finally, nonlinear Raman imaging such as SRS also needs strong Raman probes to improve the imaging quality. PDDA-based Raman probes shows not only strong Raman scattering, but also intense SRS signals. The C \equiv C bond-normalized RIE values of PDDA measured by SRS is around 200 (Figure R3C, Supplementary Fig. 7 in the Supplementary Information). We have conducted a parallel imaging study using both EdU and **P4** to demonstrate how PDDA-based probes outperform typical existing probes for SRS. As shown in Figure R6 (Supplementary Fig. 14 in the Supplementary Information), the cells stained with 50 μ M of **P4** clearly showed a good SRS imaging at a Pump/Stokes laser power setting of 10/30 mW, an integration time of 10 μ s, and based on an average of 10 frames. However, the cells stained with 200 μ M of EdU remained undetectable at this condition, and the SRS imaging on these cells could only be obtained at a much higher laser power setting (Pump/Stokes 50/100 mW), a longer integration time of 40 μ s, and based on an average of 50 frames. We have updated the manuscript and added this discussion

on Page 18 of the revised manuscript (highlighted in red).

Figure R6. SRS images of cells stained by P4 (50 μM) or EdU (200 μM) at different imaging conditions. Scale bar: 25 μm .

- 6. Strategies for polymerization are not uncommon. Appropriate articles should be cited.**

Response: Thank the reviewer for the kind suggestion. We have cited more representative references on the topochemical polymerization strategies in the revised manuscript.

- 7. The name of EdU described in line 59 is incorrect. Correctly, 5-Ethynyl-2'-deoxyuridine. Also, the structure of EdU described in Fig S1 lacks double**

bonds.

Response: We really appreciate the kind correction by the reviewer. We have corrected the name of EdU in Page 3 of the manuscript (highlighted in red). The structure of EdU in Supplementary Fig. 1 has also been amended.

Reviewer #2:

We greatly thank the reviewer for having a high opinion of the characterization work in this manuscript. We also highly appreciate the reviewer for the suggestions on the overall applications for using SRS imaging and PDDA-based Raman probes. We have carried out a series of additional experiments to address the concerns of the reviewer. Below please find our point-by-point response to these critiques.

- 1. It would be valuable to the reader to understand what the overall applications for using SRS imaging and this new probe could be and why it could be important to scientists, especially for this journal that can have a broader audience.**

Response: We really appreciate this valuable suggestion. Raman scattering microscopy has proven to be a powerful imaging method due to its many intrinsic characteristics, including capacity to provide molecular fingerprints of the sample specimen by detecting the vibrational energies associated with its chemical bonds, quantitative relationship between signal intensity and substance concentration, and narrow-band enabled multi-color imaging. Stimulated Raman scattering (SRS), as a nonlinear imaging technique that boosts vibrational excitation, represents a dramatic advance of detectability and imaging speed over conventional spontaneous Raman scattering. SRS imaging provides detailed distribution of biomolecules such as lipids and proteins. Moreover, SRS also enjoys many other advantages such as pinhole-less three-dimensional optical sectioning, non-invasive observation and deep tissue penetration, and free from non-resonant background. However, with so many advantages, the application of Raman imaging is still limited mainly because the Raman signals of the existing vibrational probes are too

weak in general. Although SRS can significantly enhance Raman signal intensity, SRS imaging on the basis of current vibrational probes still needs strong laser power, long integration time, and quite a few frames for averaging to achieve good image quality. Developing Raman probes with ultrastrong intensity, such as PDDA, is highly desired for the advancement of SRS imaging.

We have previously described some of the advantages of using Raman imaging and PDDA-based Raman probes in the first part of the manuscript. In the revised manuscript, we have added more related introduction on the overall applications of SRS imaging and the new probes on Page 3 (highlighted in red), so that the broad audience of *Nature Communications* can better understand and benefit from our work on SRS imaging and PDDA-based Raman probes.

- 2. The authors mention in their first sentence that “Bioorthogonal Raman imaging agents with enhanced Raman signals in the cellular Raman-silent region (1800–2800 cm⁻¹) are highly desired for effective Raman imaging of living cells.” But then leave the reader with no explanation as to why.**

Response: We really appreciate this kind notice. This sentence is the first sentence in the abstract, and we cannot explain too much due to word count limit. We have changed this sentence in the abstract, and updated the manuscript accordingly with the addition of the below explanation in Page 3, Paragraph 2 (highlighted in red).

“Biological molecules in a cell, such as lipids, proteins, and nuclear acids are Raman active, exhibiting strong Raman signals in the regions of 900–1800 cm⁻¹ and 2800–3100 cm⁻¹. To efficiently eliminate interference of the endogenous cellular background, Bioorthogonal Raman reporters with distinctive signals in the cell-silent vibrational region (1800–2800 cm⁻¹), such as isotopes (C-D), azides, and triple bonds (C≡C and C≡N)-containing molecules have been widely employed for labeled Raman imaging. Among the existing molecular Raman reporters, alkynes are the most promising candidates for live-cell imaging, attributed to their easy accessibility, minimal toxicity, and comparatively large Raman scattering cross-section.”

- 3. Also, how is this technique better than what is already conventionally used for instance fluorescence imaging of live cells? A little more on the applications of this new Raman probe would be helpful.**

Response: We thank the reviewer for the helpful suggestion. As we have commented in response to Q1 of the reviewer, compared with the conventional fluorescence imaging, imaging by Raman scattering microscopy has an intrinsic capacity to visualize the localization of molecules in the sample specimen. In addition, when labeled with PDDA-based Raman probes, the localized signal intensity of Raman imaging has a linear relationship with the concentration of the accumulated probe molecules, so that it can provide a quantitative information on the targeted substance. In contrast, to obtain a reliable quantitative analysis of fluorescence imaging is usually difficult due to fluorescence quenching. The PDDA-based Raman probes also enable a Raman imaging free from auto-fluorescence background and interference of endogenous cellular background.

When applied in SRS imaging, PDDA-based Raman probes allow for achieving good imaging quality and fast imaging speed at low laser power settings, so as to avoid photo damage to live cells. In addition, we presented PDDA in this manuscript as a general Raman probe platform that can be functionalized on demand. We can apply these functionalized PDDA derivatives to probe the location of specific organelles in live cells through SRS. SRS imaging also reveals the distribution of endogenous biomolecules such as lipids and proteins simultaneously through a different wavenumber region, so that we can have a more comprehensive cellular information than fluorescence imaging.

We have revised the manuscript and added according discussion on Page 19 (highlighted in red).

- 4. The biggest weakness of this paper is the cell studies. The Raman images generated aren't completely convincing with regards to the specificity of these conjugated probes to the desired targets. The fluorescence images in S9 look a**

lot more convincing than the Raman images in the end.

Response: Thank the reviewer for pointing out this issue. The same issue has been raised by the other reviewer. In our previous imaging experiments, we used FBS-free culture medium for the live cell staining to avoid possible interference of serum. However, this resulted in less healthy morphology of the cells, especially for **P4** which needed a long incubation time up to 48 h. In order to improve the quality of the live-cell SRS imaging by the PDDA-based probes, we tried different culturing conditions, including staining in FBS-containing medium to minimize the damage to live cells. As shown in updated Figure 7 in the manuscript, when the staining procedure was carried out in DMEM medium containing 10% FBS, the cell morphology was significantly improved, even for cells stained with **P4** for 48 h. This result proved that PDDA-based Raman probes were able to work in a more complicated environment than we expected before.

In our previous manuscript, since we could not conduct SRS imaging and fluorescence imaging on the same instrument, we used fluorescence imaging to characterize targeting efficiency of PDDA-based Raman probes. The precise overlap of the fluorescence of PDDA derivatives and the corresponding trackers confirmed the target specificity of these new probes, on basis of which we validated targeted organelle SRS imaging by these PDDA-based probes.

In order to further evidence the specificity of these probes to the desired targets, we used commercial fluorescence organelle trackers to co-incubate PDDA-based probes in corresponding cells, and conducted SRS imaging of PDDA-based probes and two-photon fluorescence imaging (TPFI) of commercial organelle trackers on the same instrument.

It should be noted that from a technique point of view, we can only conduct TPFI on an SRS imaging instrument, due to the limit of light source. In addition, The SRS imaging and the TPFI cannot be performed synchronously because these two imaging modes require different excitation and signal collection, and we have to run TPFI after the SRS imaging is completed. The organelles in the live cells can move when switching the imaging settings. Therefore, we cannot merge the SRS

images and the TPFPI images because the positions of the individual organelles are different. The parallel comparison between the TPFPI images of the commercial trackers and the SRS images of the PDDA-based probes clearly evidences the specificity of these conjugated probes to the desired targets (Figure R7, Supplementary Fig. 15 in the Supplementary Information). *In addition, the commercial mitochondria tracker is very weak in two-photon excitation, so that its TPFPI image is seriously interfered by the endogenous background.*

Figure R7. SRS images and two photon fluorescence images (TPFI) of HeLa cells treated with 50 μM of **P2**, **P3**, and **P4** and corresponding fluorescence organelle trackers (lysotracker red, mitotracker green, and DAPI). Scale bar: 25 μm.

5. There also needs to be more details on how long the SRS imaging took in terms of acquisition time. It would also be nice to see a control image of what an untargeted PDDA looks like after applied to the cells, to look at non-specific accumulation of the PDDA, you can't assume that the small molecule probe won't internalize passively: "PDDA itself would not be expected to accumulate

within cells, given its polyanionic nature in physiological environment”.

Response: We appreciate the kind notice. The signal integration time for a pixel of $250\text{ nm} \times 250\text{ nm}$ was $10\ \mu\text{s}$, the typical acquisition time for each frame with an area of $100\ \mu\text{m} \times 100\ \mu\text{m}$ was around 1.6 s. To further increase the signal-to-noise ratio of the image, we averaged the results of 10 frames for the final image and the total time for each image in the manuscript was less than 30 seconds. We have added this information in Methods on Page 25 and highlighted the change in red.

As the reviewer suggested, we have used pristine PDDA to stain HeLa cells for 48 h, the SRS imaging result of which is shown in Figure R8. It was almost undetectable under the same imaging condition used for the functionalized PDDA probes. The passive internalization of PDDA in live cells is less efficient than its functionalized derivatives. We have added this information in Supplementary Fig. 12.

Figure R8. SRS images of HeLa cells treated with $50\ \mu\text{M}$ of pristine PDDA for 48 h. Scale bar: $20\ \mu\text{m}$.

- 6. In Figure 7, it is unclear what the lipid channel is highlighting, just the outline of the cell? The P2 images don't look terribly impressive, why are there artifacts outside the cell region? There are also what appears to be lysosomes in the nuclear regions? And the nuclear targeting as stated in your paper is also "off" due to potential cytoplasm-nucleus translocation...but that's not good for live cell imaging applications if you can't get the real image of the nuclear**

component. Is there a way to validate the SRS images showing the presumed targeting of the PDDA to your targets with fluorescence images at the same time? To overlay their distribution within the cell? Again, it would be nice to know how long it took to generate these SRS images.

Response: We thank the reviewer again for the very valuable comments. As we have mentioned in our response to Q4 of the reviewer, the reason for the poor quality of our previous cell images was that we used FBS-free culture medium. After we changed to FBS-containing culture medium, the quality of SRS imaging has been significantly improved. As shown in updated Figure 7 in the manuscript, we used the 2850 cm^{-1} channel to highlight the cell morphology and the 2120 cm^{-1} channel to highlight the targeting probes. There were no artifacts outside the cell region any more. In addition, the imaging results of lysosomes, mitochondria, and nuclei became much cleaner than before. We have also used commercial organelle trackers to validate the targeting of PDDA-based probes, and the results and discussion can be found in our response to Q4 and Figure R7.

- 7. Clearly the PDDA has a nice peak in the “silent region”, and it seems to outperform many of the pre-existing probes, but using it in this live cell imaging application doesn’t seem all that convincing as of yet. I believe more details on how it outperforms conventional live cell imaging (ie. fluorescence) and more cellular imaging experiments are necessary before publication under the current title “Polydiacetylene-Based Ultrastrong Bioorthogonal Raman Probes for Targeted Live-Cell Raman Imaging”.**

Response: We highly appreciate the reviewer for raising up these valuable comments and suggestions. In terms of Raman intensity, PDDA and its derivatives have outperformed most pre-existing Raman probes with ultrastrong Raman signals in the cellular “silent region”. We agree that in our previous work, the performance of PDDA-based Raman probes was not convincing. However, after we have modified our staining protocol, the newly obtained SRS imaging on the basis of these probes unambiguously excelled the existing Raman probes to a large extent.

We have conducted a parallel imaging study using both EdU and **P4** to demonstrate how PDDA-based probes outperform typical existing probes for SRS. As shown in Figure R9 (same as Figure R6), while the cells stained with **P4** ($50 \mu\text{M}$) clearly showed a good SRS imaging at a Pump/Stokes laser power of 10/30 mW, an integration time of $10 \mu\text{s}$, and with an average of 10 frames, the cells stained with EdU ($200 \mu\text{M}$) remained undetectable at this condition. The SRS imaging on these cells could only be obtained at much higher laser powers (Pump/Stokes 50/100 mW), longer integration time ($40 \mu\text{s}$) and larger number of frames for averaging (50 vs 10). This discussion has been added in the manuscript on Page 18 (highlighted in red).

Figure R9. SRS images of cells stained by **P4** ($50 \mu\text{M}$) or EdU ($200 \mu\text{M}$) at different imaging conditions. Scale bar: $25 \mu\text{m}$.

In terms of the comparison between SRS imaging with PDDA-based Raman probes and the conventional fluorescence imaging, we have discussed in details above in our response to Q1 and Q3 of the reviewer. Compared with fluorescence imaging of live cells, Raman imaging, particularly SRS imaging, holds many advantages such as molecular fingerprint information, linear signal intensity dependence on sample concentration, free from endogenous fluorescence background, etc. The development of PDDA-based probes has successfully addressed intensity problem of existing Raman probes. High quality SRS imaging can be achieved at low laser power and short acquisition time when applying PDDA-based probes. In addition, as the reviewer suggested, we have carried out extensive cell-based imaging experiments to demonstrate targeted live-cell Raman imaging on the basis of PDDA derivatives. With the development of more Raman probes with improved performance such as PDDA, and the advancement of optical imaging techniques, Raman imaging is expected to break the current intensity bottleneck and achieve broader biomedical applications shortly.

8. Minor: The last sentence of the conclusion also needs to be restructured grammatically.

Response: Thanks for the suggestion. We have changed the statement of the last sentence accordingly.

References

- (1) Hirakawa, A. Y.; Tsuboi, M.: Molecular Geometry in an Excited Electronic State and a Preresonance Raman Effect. *Science* **1975**, 188, 359-361.
- (2) Wei, L.; Min, W.: Electronic Preresonance Stimulated Raman Scattering Microscopy. *The Journal of Physical Chemistry Letters* **2018**, 9, 4294-4301.
- (3) Brown, A. J.; Rumbles, G.; Phillips, D.; Bloor, D.: Sidegroup dependence of chromism in polydiacetylenes. *Chemical Physics Letters* **1988**, 151, 247-252.
- (4) Okada, S.; Peng, S.; Spevak, W.; Charych, D.: Color and chromism of

polydiacetylene vesicles. *Accounts of Chemical Research* **1998**, 31, 229-239.

(5) Batchelder, D.: Colour and chromism of conjugated polymers. *Contemporary Physics* **1988**, 29, 3-31.

Reviewers' Comments:

Reviewer #1:

Remarks to the Author:

The authors responded politely and appropriately to all comments from reviewers and the current manuscript would be of interest to readers. Therefore, reviewer recommend for publication in Nature Communications.

Reviewer #2:

Remarks to the Author:

The authors have spent considerable effort in revising the manuscript. They have also included new cell experiments that show better specificity of their new imaging agent to various cellular regions.

They demonstrate that their new Raman probe has 10^4 more Raman signal than other alkynes developed in the silent region which is of great interest.

The author's response to my question about how this method has advantages over conventional fluorescence imaging is not entirely convincing. Cellular and molecular biologists continue to use fluorescence imaging for cell studies even though these probes have photobleaching and autofluorescence properties. In fact, more recently several groups are using the intrinsic autofluorescence of cellular features to create label-free fluorescent images of cells.

The real power of Raman imaging has always been its ability to multiplex beyond fluorescence. In this case the authors are presenting a single probe with a Raman signal in the "silent region". Although it appears to be more sensitive it is still in fact just a single probe with a distinct signature that doesn't appear to have multiplexing properties.

In Suppl Figure 13 the authors attempt to compare their staining approach to conventional fluorescence cellular imaging with multiple fluorescent dyes. The only problem is that their new probe can only stain and report one particular part of the cell at one time, whereas the multiple fluorophores allow for simultaneous staining and imaging. I understand that they intend to use the intrinsic Raman signal from the cell's various features (lipids, DNA, proteins, etc.), but in the end they can only target/track a single event with their new probe. The probe does appear to be more sensitive than existing Raman probes in this silent region and I suppose that is the most impressive part of this new imaging approach.